# Mild Endurance Exercise during Fasting Increases Gastrocnemius Muscle and Prefrontal Cortex Thyroid Hormone Levels through Differential BHB and BCAA-Mediated BDNF-mTOR Signaling in Rats

**DOI:** 10.3390/nu14061166

**Published:** 2022-03-10

**Authors:** Antonia Giacco, Federica Cioffi, Arianna Cuomo, Roberta Simiele, Rosalba Senese, Elena Silvestri, Angela Amoresano, Carolina Fontanarosa, Giuseppe Petito, Maria Moreno, Antonia Lanni, Assunta Lombardi, Pieter de Lange

**Affiliations:** 1Dipartimento di Scienze e Tecnologie, Università Degli Studi del Sannio, Via De Sanctis, 82100 Benevento, Italy; agiacco@unisannio.it (A.G.); federica.cioffi@unisannio.it (F.C.); elena.silvestri@unisannio.it (E.S.); moreno@unisannio.it (M.M.); 2Dipartimento di Scienze e Tecnologie Ambientali, Biologiche e Farmaceutiche, Università Degli Studi Della Campania “Luigi Vanvitelli”, Via Vivaldi 43, 81130 Caserta, Italy; arianna.cuomo@unicampania.it (A.C.); roberta.simiele@unicampania.it (R.S.); rosalba.senese@unicampania.it (R.S.); giuseppe.petito@unicampania.it (G.P.); antonia.lanni@unicampania.it (A.L.); 3Dipartimento di Scienze Chimiche, Università Degli Studi di Napoli “Federico II”, Monte Sant’Angelo, Via Cinthia 4, 80126 Naples, Italy; angela.amoresano@unina.it (A.A.); carolina.fontanarosa@unina.it (C.F.); 4Istituto Nazionale Biostrutture e Biosistemi, Viale delle Medaglie d’Oro 305, 00136 Rome, Italy; 5Dipartimento di Biologia, Università Degli Studi di Napoli “Federico II”, Monte Sant’Angelo, Via Cinthia 4, 80126 Naples, Italy

**Keywords:** tissue thyroid hormone (T3), mild endurance exercise, fasting, tissue BHB, tissue BCAA

## Abstract

Mild endurance exercise has been shown to compensate for declined muscle quality and may positively affect the brain under conditions of energy restriction. Whether this involves brain-derived neurotrophic factor (BDNF) and mammalian target of rapamycin (mTOR) activation in relation to central and peripheral tissue levels of associated factors such as beta hydroxy butyrate (BHB), branched-chain amino acids (BCAA) and thyroid hormone (T3) has not been studied. Thus, a subset of male Wistar rats housed at thermoneutrality that were fed or fasted was submitted to 30-min-mild treadmill exercise bouts (five in total, twice daily, 15 m/min, 0° inclination) over a period of 66 h. Prefrontal cortex and gastrocnemius muscle BHB, BCAA, and thyroid hormone were measured by LC-MS/MS analysis and were related to BDNF and mammalian target of rapamycin (mTOR) signaling. In gastrocnemius muscle, mild endurance exercise during fasting maintained the fasting-induced elevated BHB levels and BDNF-CREB activity and unlocked the downstream Akt-mTORC1 pathway associated with increased tissue BCAA. Consequently, deiodinase 3 mRNA levels decreased whereas increased phosphorylation of the mTORC2 target FOXO1 was associated with increased deiodinase 2 mRNA levels, accounting for the increased T3 tissue levels. These events were related to increased expression of CREB and T3 target genes beneficial for muscle quality previously observed in this condition. In rat L6 myoblasts, BHB directly induced BDNF transcription and maturation. Mild endurance exercise during fasting did not increase prefrontal cortex BHB levels nor was BDNF activated, whereas increased leucine levels were associated with Akt-independent increased phosphorylation of the mTORC1 target P70S6K. The associated increased T3 levels modulated the expression of known T3-target genes involved in brain tissue maintenance. Our observation that mild endurance exercise modulates BDNF, mTOR and T3 during fasting provides molecular clues to explain the observed beneficial effects of mild endurance exercise in settings of energy restriction.

## 1. Introduction

Weight loss by nutritional interventions often leads to muscle weakness [1]. In studies focusing on evaluating optimal weight-loss strategies, mild endurance exercise has emerged to be sufficient to counteract the negative impact of various energy-restricting nutritional interventions on muscle strength and quality in obese and non-obese subjects (see, for review, [1]), and in rodents alternate fasting and exercise has likewise been shown to improve endurance performance [2]. In this light, at the biochemical level there exists evidence for a compensatory role for exercise on fasting-mediated protein breakdown linked to insulin-independent reactivation of Akt/mammalian target of rapamycin (mTOR) C1 signaling in mice [3] and humans [4], which would impact muscle mass and quality. We have recently shown that a short protocol including a 3-day fasting period combined with submaximal daily cycling ameliorates human body composition by increasing the lean (muscle) mass/fat mass ratio [5]. A similar protocol carried out under thermoneutral conditions in the rat led to drastic biochemical changes in the gut, liver and skeletal muscle, the latter tissue showing increased expression of BDNF [6], a factor involved in neuroprotection that has been studied extensively in relation to exercise in the brain [7,8,9]. 

With regard to the brain, BDNF signaling has been shown to be activated by acute exercise both in the cortex [7,8] and hippocampus, improving spatial learning and memory [8,9]. It has recently been shown that prolonged exercise promotes hippocampal expression of BDNF induced by beta hydroxy butyrate (BHB) [10]. BDNF has been shown to induce mammalian target of rapamycin (mTOR)C1 activity in response to energetic challenges in the cortex, with beneficial effects for cognition [11]. 

A link between skeletal muscle BHB-related BDNF and mTOR signaling in response to energy restriction and mild exercise has not been studied, although muscle-specific BDNF ablation has been shown to block metabolic switching to lipid metabolism in fasted mice, which was observed only in females [12]. Several nutrients, such as branched-chain amino acids (BCAAs), are known to stimulate muscle protein synthesis via mTOR activation [13,14]. Although in mice exercise did not increase fasting serum levels of BCAA including leucine [2,3], increased protein synthesis through mTOR activation during fasting in rat gastrocnemius and plantaris muscle has been shown to be inducible 1h after exogenous administration of leucine [15]. This result brings to light the importance of measuring tissue BCAA levels in response to exercise during fasting to uncover their role in protein synthesis. 

In this context, one key factor, namely thyroid hormone, impacts central and peripheral physiological events ranging from muscle mass and function to behavior [16]. Indeed, hypothyroidism is associated with muscle weakness (see, for review, [17]). Physical exercise is one condition that may influence T3 serum and tissue availability. Notably, the metabolic effects of exogenous administration of thyroid hormones share several similarities with those elicited by exercise [18]. The upregulation of free (f) T3 and fT4 serum levels by low-intensity endurance exercise has been observed in humans [19,20] and is associated with amelioration of academic performance in one study [20], and exercise intensity positively correlates with fT3 serum levels [19]. However, T3 serum levels are not consistently shown to be associated with physical activity, neither in humans [21] nor in rodents [22,23]. 

T3 tissue levels that derive from serum T3 and T4 are the result of the concerted action of specific transporters and deiodinases under varying physiological conditions [24]. A crucial role for T3 in the response to exercise, both centrally and peripherally, has been highlighted based on studies using deiodinase 2 knockout mice [23,25]. Because deiodinase expression has been shown to be modulated by mTOR signaling towards increased T3 synthesis [26,27], it is conceivable that increased activity of this pathway would result in increased local T3 levels. Since BHB, BCAA, and T3 tissue levels in response to exercise or exercise with fasting in target tissues have not been studied, we investigated whether mild endurance exercise during fasting modulated BDNF-mTOR signaling in relation to BHB and BCAA tissue levels, and whether this would alter T3 tissue levels using male Wistar rats. The low exercise intensity was chosen with the aim to minimalize the effects of the intervention per se while inducing biochemical pathways related to the preservation of muscle mass when combined with fasting, a phenotype that emerges from similar interventions in human subjects [1]. 

We studied the response to mild endurance exercise during fasting in both gastrocnemius muscle and prefrontal cortex of rats to obtain clues as to how maintenance of muscle mass and strength is achieved, and whether this intervention would beneficially affect the prefrontal cortex.

## 2. Materials and Methods

### 2.1. Materials

The primary antibodies against phospho-Akt (Ser473), pan-Akt, Phospo-CREB (Ser133), pan-CREB, phospho-TRKA (Tyr 674-675)/TRKB(Tyr706-707), pan-TRKB, phospho 4EBP1 (Thr37/Thr46), pan 4EBP1, phospho FOXO1 (Ser256), pan FOXO1, phospho P70S6K (Thr389), pan P70S6K, and Caspase 3, were obtained from Cell Signaling Technology (Beverly, MA, USA). BDNF was from Santa Cruz Biotechnology (Dallas, TX, USA). phospho-TRKB (Tyr816) was purchased from Millipore (Burlington, MA, USA). The β-actin antibody was from Bioss Antibodies Inc. (Woburn, MA, USA). The appropriate horseradish peroxidase-conjugated secondary antibodies were purchased from Abcam (Cambridge, UK). For Q-PCR, Power SYBR Green PCR Master Mix and 96-well optical reaction plates were from Applied Biosystems (Thermo Fisher Scientific, Waltham, MA, USA). The primers were synthesized and sequenced by Eurofins (Milan, Italy). Beta Hydroxy Butyrate was purchased from Sigma Aldrich (Merck kGaA, Darmstadt, Germany). All standards for mass spectrometry were purchased from Sigma-Aldrich S.r.l. (Milan, Italy). All the solutions and solvents were of the highest available purity and were suitable for LC–MS analysis and purchased from J. T. Baker (Phillipsburg, NJ, USA).

### 2.2. Animal Handling

Male Wistar rats (*n* = 16, 12 weeks old, weight approximately 300 g), housed separately at thermoneutrality (28 °C), had ad libitum access to water and chow [Fatty acid content (mg/kg): palmitate (16:0) 4387; palmitoleate (16:1) 202; stearic (18:0) 675; oleic (18:1) 5046; linoleic (18:2) 12335; linolenic (18:3) 1169. Total metabolizable percentage of energy: carbohydrates 60.4; proteins 29; fat 10.6; 15.88 KJ gross energy/g (Muscedola s.r.l., Milan, Italy)]. During the 3-week-acclimation to the housing temperature, rats were familiarized with the treadmill (Panlab, Harvard Apparatus, Holliston, MA, USA). There were four experimental conditions used, each with a total of four rats used per condition. The first consisted of ad libitum feeding (control group, C), the second of ad libitum feeding and submission to exercise (exercised group, E), the third of submission to fasting (fasted group, F) and the fourth of submission to fasting and to exercise (fasted and exercised group, FE). All rats had ad libitum access to water. Food was withdrawn for a period of 66 h. The exercise program consisted of five low-intensity treadmill runs (twice daily, 30 min, 15 m/min, 0° inclination). All rats were familiarized with the treadmill, the initial environmental temperature was set at 25 °C and constantly monitored to ensure that the temperature inside the plexiglass cover did not exceed 28 °C during the exercise sessions. After resting on the treadmill for 15 min, rats were placed back in their cages at 28 °C. The duration of the experiment was 66 h (from the start of fasting to sacrifice). A timeline of the experimental setup and body weight, food intake, and metabolic parameters were published elsewhere [6]. Exercised rats were sacrificed 4 h after completion of the last exercise bout. After sacrifice serum was collected and prefrontal cortex and gastrocnemius muscles were quickly removed, immediately frozen in liquid nitrogen, and stored at −80 °C. All rats were analyzed applying each technique used in this study and for each sample triplicate technical repetitions were performed. All animal experiments comply with the ARRIVE guidelines and have been carried out in accordance with EU Directive 2010/63/EU for animal experiments. All protocols were approved by the University of Campania “Luigi Vanvitelli” Ethics of Animal Experiments Committee and the Italian Ministry of Health (authorization 704/2016 PR, article 31 legislative decree 26/2014).

### 2.3. Cell Culture and Treatment

Rat L6C5 myoblasts (a subline of the ATTC line CRL 1458) were obtained from the Cell Bank (Interlab Cell Line Collection) of the National Institute for Cancer Research (Genoa, Italy), and cultured at early passage numbers in DMEM containing 10% fetal bovine serum (Hyclone, Logan, UT, USA). Cells were grown to 75% confluence in DMEM containing 10% fetal bovine serum (Hyclone), then treated with Beta Hydroxy Butyrate (8 mM) in DMEM with or without glucose (25 mM) and 10% fetal bovine serum for 3 days. Cells were lysed for RNA or protein isolation and were further processed for Q-PCR or Western blotting.

### 2.4. Quantitative Analysis of Serum and Tissue T3 (Triiodothyronine), T4 (Thyroxine), and Tissue BHB (Beta-Hydroxy Butyric Acid), and BCAA (Branched Chain Amino Acids)

Sample treatment: 100 µL serum, 5 mg prefrontal cortex, or 10 mg gastrocnemius muscle was digested with 20 µL of elastase (1 mg/mL) + 80 µL AMBIC 50 mm O.N 37 °C. To the digested sample 100 µL Methanol −20 °C O.N. was added. The samples were centrifugated and the supernatant was dried in a savant to be resuspended in 50 µL of Methanol and analyzed by LC-MS/MS in MRM mode. Quantitative analysis of T3 (triiodothyronine), T4 (thyroxine), BHB (beta-hydroxy butyric acid), and branched-chain amino acids (BCAA) was performed by means of liquid chromatography-tandem quadrupole mass spectrometry (LC–MS/MS) in MRM mode. LC–MS/MS instrumentation and conditions: 1 μL of supernatant were analyzed by using a 6420 triple quadrupole system with an HPLC 1100 series binary pump from Agilent Technologies. The analytical column was a Phenomenex Kinetex 5u 100 A C18. The mobile phase was generated by mixing eluent A (H_2_O 0.1% Formic Acid) and eluent B (95% ACN and 0.1% Formic Acid) and the flow rate was 0.200 mL/min. Starting condition was 5% to 95% A for 8 min, then to 100% for 2 min. Tandem mass spectrometry was performed using a turbo ion spray source operated in positive ion mode, and the MRM mode was used for the selected analytes.

T3, T4, BHB and BCAA standard solutions were prepared by dissolving the equivalent of 1 mg of the respective metabolites in 1 mL of methanol. standard solution of 500 pg\µL of each metabolite was used for the optimization of the MRM transition. Metabolites were automatically (flow injection) tuned for ionization polarity, product ion, and collision energy (CE) using metabolite standard solutions via Agilent MassHunter Optimizer software. Table 1 provides a list of precursor ions, product ions, collision energy and polarity.

Data Processing: Extracted mass chromatogram peaks of metabolites were integrated using Agilent MassHunter Quantitative Analysis software (B.05.00). Peak areas of corresponding metabolites are then used, as quantitative measurements, for assay performance assessments such as assay variation and linearity. With regard to quantification of analytes, the first step for the setting of mass spectral analysis consisted in the MRM detection of the analytes individually infused to establish the optimal instrument settings for each compound. Standard calibration curves were constructed by plotting peak areas against concentration (pg/µL), and linear functions were applied to the calibration curves. Data were integrated by Mass Hunter quantitative software showing a linear trend in the calibration range for all molecules. The coefficients of determination (R2) were greater than 0.99 for all analytes. Once set up, the MRM mass spectral method was applied to the analyses of specific sets of serum samples. The stock standard solutions were prepared by adding 1.00 mL aliquots of each analyte to a 10-mL volumetric flask and bringing the standard to volume with methanol to yield a standard solution with 1000 µg/L of each analyte. The stock solutions were stored at −20 °C until the analysis. Standard mixtures for each analyte were prepared by serious dilution as follows: 1–5–10–25–50 pg/uL. MassHunter Quantitative Analysis software (B.05.00) was used for data analysis. Table 1 shows the relevant details of the compounds under study.

### 2.5. Gene Expression Analysis

Total RNA was extracted from frozen tissues using TRIZOL reagent (Invitrogen). cDNA was synthesized from 1 µg of RNA in 20 µL reaction volume, according to Quanti Tect Reverse Transcription Kit instruction (Qiagen, Hilden, Germany). Amplification mixes for QPCR were carried out with 50 nM gene-specific primers and Power SYBR Green PCR Master Mix on a Quant Studio 5 real-time PCR machine (Applied Biosystems, Thermo Fisher Scientific, Waltham, MA, USA) using standard cycle parameters. The 2−∆∆Ct method was used to calculate relative gene expression changes normalized to the expression of the housekeeping gene, βeta-actin. For primers used for gene expression analysis, see Table 2.

### 2.6. Western Immunoblot Analysis

Protein and phosphoprotein levels were determined by immunoblotting using total lysates from rat tissue or L6 cells by employing the above-described specific antibodies. β actin was used as a loading control. The ratio of the phosphorylated versus total proteins was calculated to assess the phosphorylation status.

### 2.7. Statistical Analysis

The sample size was deduced by a Power test using the G*Power software version 3.1.9.2 from the Heinrich Heine University of Dusseldorf (http://www.gpower.hhu.de, accessed on 13 April 2020). The power analysis was based on a preliminary experiment and revealed that to compare four groups and observe expected effects within the physiological range, with α set at 0.05 and the power at 0.90, resulting in an effect size of 1.27, the number of rats to be used in each sample is four. This implies that samples obtained from four rats from each group are sufficient to observe statistically significant differences with the parameters set as such using one way ANOVA (post-hoc test: Newman-Keuls). Data are expressed as means ± SEM. Statistical differences were analyzed by one way ANOVA followed by Newman-Keuls post-hoc testing. Statistical analysis was performed by using Prism 5.0 (Graphpad, San Diego, CA, USA). Differences between treatments were considered significant at *p* < 0.05.

Absolute ANOVA P values are listed as Appendix A.

## 3. Results

### 3.1. Gastrocnemius Muscle BHB and BCAA Levels in Relation to BDNF-mTORC1 Signaling in Response to Mild Endurance Exercise, Fasting, and Their Combination 

We first investigated whether BHB and BCAA are involved in the response to mild endurance exercise during fasting in gastrocnemius muscle, and how this could be related to BDNF-mTOR signaling. Both mild endurance exercise and fasting increased gastrocnemius muscle BHB 1.4-fold and fasting with exercise nearly doubled BHB levels (1.8-fold increase, see Figure 1A). There was no difference in mature BDNF in chow-fed exercised rats compared to controls (Figure 1B, left panel). (TRKB) Tyr706/707 phosphorylation was not significantly changed in chow-fed exercised rats compared to controls (Figure 1B, left panel), neither was phosphorylation of CREB (Ser133) (Figure 1B left panel) or of Akt (Ser473). In addition, phosphorylation of mTORC1 targets P70S6K(Thr389), 4EBP1 (Thr37/Thr46) remained unaltered despite a modest increase in BCAA (1.3-fold for leucine, 1.7-fold for isoleucine, and 1.4-fold for valine) whereas increased phosphorylation of the mTORC2 target FOXO1 (Ser256) was accompanied by increased protein levels (Figure 1B central panel). These data show that the chosen exercise intensity did not significantly affect BDNF and mTOR signaling. With respect to the controls and chow-fed exercised rats, prolonged exposure to increased BHB during fasting resulted in a consistent 3-fold increase in BDNF maturation as well as in CREB (Ser133) and TRKB Tyr706/707 phosphorylation, irrespective of whether rats were subjected to the exercise bouts (Figure 1B, left panel). Phosphorylation levels of Akt (Ser473) and mTORC2-Akt target FOXO1 (Ser256), as well as the mTORC1 targets 4EBP1 (Thr37/Thr46) and P70S6K (Thr389) were all reduced by fasting (down to 0.3-fold (pAkt) and 0.5-fold the control levels, respectively) but did not differ significantly from control levels when fasting was combined with mild endurance exercise (Figure 1B, central panel). Of note, P70S6K protein levels were also strongly downregulated by fasting in gastrocnemius muscle, and to a lesser extent also those of Akt and 4EBP1 (Figure 1B, central panel). In conjunction with the above data, with respect to controls, mild endurance exercise combined with fasting induced the gastrocnemius muscle levels of the BCAA Valine 1.3-fold, and the levels for leucine and isoleucine 3-fold (Figure 1A). To assess whether the interventions caused induction of apoptosis, we measured muscle Caspase3 cleavage which increased two-fold with mild endurance exercise with respect to controls (Figure 1B, right panel), nevertheless, the ratio cleaved Caspase3/Procaspase3 remained below 0.1 (arbitrary units) which indicates non-significant induction of apoptosis [28]. 

### 3.2. Gastrocnemius Muscle T3 and T4 Levels in Relation to Deiodinase, and BHB and Amino acid Transporter Expression in Response to Mild Endurance Exercise, Fasting, and Their Combination

Next, we analyzed whether the different interventions caused alteration of gastrocnemius muscle levels of T3 and T4. Mild endurance exercise increased T3 levels 1.3-fold, fasting caused no change, and combined with fasting increased T3 levels by 1.6-fold with respect to controls (Figure 2A). In contrast, serum (total) T3 levels were decreased both by mild endurance exercise and fasting, to 75% and 50%, respectively, whereas the combined intervention caused no change (Figure 2A). Gastrocnemius muscle T4 levels did not vary between conditions. Serum total T4 levels decreased to 67% of those of the control values during fasting (Figure 2A). In conjunction with the changes in T3 levels, with respect to controls, gastrocnemius muscle deiodinase 2 mRNA levels were increased 2-fold by mild endurance exercise and 2.5-fold in combination with fasting (Figure 2B), whereas deiodinase 3 mRNA levels were increased 2-fold by mild endurance exercise, and 6.5-fold by fasting, an induction that was brought down to 3.5-fold when the interventions were combined (Figure 2B). With respect to controls, mRNA levels for the high-affinity BHB transporter MCT2 significantly increased being 3.5-, 2.5- and 3.0-fold in response to exercise, fasting and their combination, respectively, in line with the increased tissue BHB levels (Figure 1A). With respect to controls and mild endurance exercise alone, fasting increased mRNA levels of the high-affinity leucine transporter SLC7A5 (LAT1) 2-fold, and the combined intervention by 2.5-fold. Increased mRNA levels of the amino acid transporter SLC7A8 (LAT2) were only observed in the fasting condition (2.5-fold) (Figure 2B).

### 3.3. Effects of BHB on BDNF Transcription, Translation, and Maturation in L6 Cells

In light of the observation that mature BDNF levels conditionally correlated with increased BHB in gastrocnemius muscle, using rat L6 muscle cells, we set out to explore whether the link between these parameters was causal or non-functional. To do so, we studied the effect of BHB either in the presence or absence of 25 mM glucose. The chosen glucose concentrations were not based on physiological serum levels but rather on glucose levels within the muscle. We reasoned that a high glucose concentration would mimic glucose levels in the working muscle during exercise that increase up to 4-fold [29], and the absence of glucose in the presence of full serum would mimic intracellular glucose in glycogen depleted muscle during prolonged fasting. A subset of cells was incubated with 8 mM BHB. Cell morphology was unaltered following treatment with BHB (Figure 3A). In the presence of glucose, BHB induced a 2.5-fold increase in BDNF mRNA levels (Figure 3B). In line with what is observed in the fasted gastrocnemius muscle [6], glucose deprivation caused a 3.5-fold upregulation of BDNF mRNA levels and BHB additionally upregulated BDNF mRNA to 5.5-fold. In the presence of glucose, BHB caused a 3-fold increase in BDNF maturation, but in the absence of glucose mature BDNF levels were not detected in each condition (Figure 3C). Although these results show that the studied interventions in muscle cells do not fully reflect the in vivo situation, they also demonstrate that BHB induces BDNF transcription and maturation in the muscle cell, supporting a functional role for BHB in BDNF regulation in vivo.

### 3.4. Prefrontal Cortex BHB and BCAA Levels in Relation to BDNF-mTORC1 Signaling in Response to Mild Endurance Exercise, Fasting, and Their Combination

Since the prefrontal cortex is a known target of BHB-induced BDNF activation by exercise, we subsequently studied the effect of the experimental interventions in this tissue. Perhaps surprisingly, prefrontal cortex BHB levels were low and did not vary between conditions (Figure 4A). Despite a lack of increase in BHB levels and a non-significant overall variation of BDNF maturation (Figure 4B, left panel), with respect to sedentary fed controls, mild endurance exercise caused activation of BDNF-mTORC1 signaling in the prefrontal cortex. CREB phosphorylation (Ser133) increased 4-fold, TRK A (Tyr 646) and B (Tyr 706/707) increased 2-fold, and TRKB (Tyr 816) increased by 1.5-fold (Figure 4B. left panel). Phosphorylation of Akt (Ser473) and the mTORC1 target 4EBP1 (Thr37/Thr46) both increased 1.5-fold (Figure 4B, right panel). The same result was obtained for the mTORC1 target P70S6K (Thr389), and this protein was also phosphorylated during fasting either in combination with mild endurance exercise or not, conditions during which Akt phosphorylation was not induced (Figure 4B, right panel). With respect to controls, prefrontal cortex leucine levels increased 1.3-fold during exercise and 1.9-fold during fasting and increased 1.4-fold in response to the combined intervention. Isoleucine and valine levels did not vary (Figure 4A). The observed changes in leucine levels did not correlate with the observed changes in BDNF signaling, but induced leucine levels did correlate with Akt-independent induction of P70S6K phosphorylation. Taken together, these results indicate that although mild endurance exercise induces signaling downstream of BDNF, in combination with fasting this exercise intensity does not affect prefrontal cortex BDNF-TRKB-Akt signaling.

### 3.5. Prefrontal Cortex T3 and T4 Levels in Relation to Deiodinase, and BHB and Amino Acid Transporter Expression in Response to Mild Endurance Exercise, Fasting, and Their Combination

In the prefrontal cortex, with respect to controls, T3 levels in response to mild endurance exercise were reduced to 0.6-fold (Figure 5A) following the serum levels (Figure 2A) but did not change during fasting (Figure 5A) despite the decline in serum levels (Figure 2A), whereas the combined intervention caused a significant 1.3-fold increase (Figure 5A) which followed the increase in serum levels. Prefrontal cortex T4 levels failed to be detected accurately with our method. Neither mild endurance exercise alone nor combined with fasting induced deiodinase 2 mRNA levels in the prefrontal cortex (Figure 5B). Mild endurance exercise decreased deiodinase 3 mRNA levels to 0.5-fold those of the controls, whereas, with respect to all other conditions, during fasting mRNA levels were significantly brought down to 0.3-fold, being brought back to 0.5-fold in combination with mild endurance exercise (Figure 5B). In line with the lack of variation of BHB levels between each intervention (Figure 4A), prefrontal cortex mRNA levels for the high-affinity BHB transporter MCT2 showed no change (Figure 5B). With respect to controls, mRNA levels for the large neutral amino acids transporter with high affinity for leucin (LAT)1 increased 2.5-fold by the combined intervention, whereas those for LAT2 remained unaltered in response to each intervention.

### 3.6. Effects of Mild Endurance Exercise, Fasting, and Their Combination on the Expression of Prefrontal Cortex Genes Involved in Thyroid Hormone Transport and Ketone Metabolism, and BDNF, TRKB, and PGC-1α

With regard to thyroid hormone transporters, only MCT10 mRNA levels were lowered with respect to controls by mild endurance exercise (0.25-fold), fasting (0.4-fold), and fasting with exercise (0.3-fold). MCT8 and OATP1C1 mRNA was unaltered in each condition (Figure 6A). Regarding genes encoding ketolytic enzymes, with respect to controls, there was a significant but modest decrease in mRNA levels for OXCT1 by mild endurance exercise (0.75-fold), fasting (0.5-fold) and fasting with exercise (0.75-fold), those for BDH1 did not change (Figure 6A). BDNF, TRKB, and PGC-1α mRNA levels remained unaltered (Figure 6A). 

### 3.7. Effects of Mild Endurance Exercise, Fasting, and their Combination on the Expression of Known Prefrontal Cortex T3 Responsive Genes

Next, to assess the effect of the increased levels of T3 in the prefrontal cortex induced by mild endurance exercise during fasting, we studied the transcriptional effect of six known cortex T3 target genes identified by transcriptome analysis [30,31] termed lysin-specific demethylase (HR), semaphorin 3c (SEMA3C), RASD family, member 2 (RASD2), krüppel-like factor 9 (KLF9), sonic hedgehog (SHH), and calbindin (CALB1). The mRNA levels of these genes were all modulated by the combined intervention (Figure 6B), which is in accordance with the increase in T3 levels with respect to fasting alone in this condition (Figure 5A). The combined intervention increased mRNA levels of KLF9 and HR (1.5-fold) and RASD2 (2-fold) with respect to all other conditions. The 1.5-fold increase in SHH and SEMA3C mRNA levels reached significance with respect to mild endurance exercise and controls, and with fasting and controls, respectively. With respect to fasting alone, the combination with mild endurance exercise significantly decreased the expression of CALB1, a gene of which transcription is repressed by T3 (reaching a level of 0.75-fold with respect to controls, fasting levels being 1.2-fold those of controls) (Figure 6B).

## 4. Discussion

The present study highlights a functional association between the effect of mild endurance exercise during fasting on tissue BHB and BCAA-mediated BDNF-mTOR signaling and gastrocnemius muscle and prefrontal cortex T3 levels. Fasting-related increased gastrocnemius muscle BDNF signaling can be functionally coupled to the chronic increase in gastrocnemius muscle levels of BHB since we observed that this compound directly stimulates BDNF activation in rat L6 myoblasts, in analogy to what has been described in cortical neurons [32]. Since rat L6 cells lack the TRKB receptor [33], downstream effects of BHB-BDNF activation were not studied. Our study reveals that when combined with fasting, mild endurance exercise can result in activation of mTOR signaling in response to tissue BCAA levels including leucine, which does not occur under fed conditions. In line with our results, gastrocnemius muscle sensitivity to leucine-dependent mTOR activation has very recently been shown to occur only through intense resistance exercise under fed conditions [34]. Supporting evidence for a functional association between the observed increased tissue BCAA levels, especially those of leucine and isoleucine, and re-activation of gastrocnemius muscle mTOR signaling comes from a study showing that exogenous leucine administration during fasting rapidly induces mTOR signaling in rat gastrocnemius muscle [15]. Increased gastrocnemius muscle T3 levels in response to mild endurance exercise during fasting are mediated by the combination of increased expression of deiodinase 2, an enzyme that increases T3 synthesis, and decreased expression of deiodinase 3, an enzyme that reduces the T3 pool. These events can be associated with intact BDNF-Akt-mTOR signaling. Evidence for this association comes from studies in cell systems showing that inhibition of mTORC1 activity increases expression of deiodinase 3 [26] and that mTORC2-Akt mediated phosphorylation and inactivation of FOXO1 induces transcription of deiodinase 2 [27]. 

Of note, although BDNF and mTOR pathways have been shown to intertwine through BDNF-induced Akt activation in response to energetic challenges in the brain [11], we observed that gastrocnemius muscle BDNF-CREB activation during fasting is, as is expected in this tissue, not coupled with Akt-mTOR activation. In addition, and perhaps in contrast to what one would expect, we did not observe a significant decrease in T3 levels in gastrocnemius muscle of fasted rats with respect to control rats, despite increased deiodinase 3 mRNA levels. One explanation for this may come from our previous observation in the same model that gastrocnemius muscle mRNA levels of MCT8 and MCT10, both involved in T4 and T3 transport, significantly increased during fasting [6]. Combined with the observed increased expression during fasting of LAT2, a transporter with an affinity for leucine as well as for T3 [35], these findings may reflect the need for maintenance of T3 levels in muscle to safeguard strength since protein synthesis is not stimulated in this condition. The persisting increased MCT8 and MCT10 mRNA levels when fasting was combined with exercise [6] provide an additional explanation for the observed increased gastrocnemius muscle T3 levels in this condition. 

Data from our previous study in the same animal model [6] on the expression of key genes in gastrocnemius muscle in response to mild endurance exercise during fasting can be related both to increased mTOR signaling-induced tissue T3 levels and increased BDNF-CREB activity. 

The increased gastrocnemius muscle T3 levels can be related to increased uncoupling protein 3 (UCP3) and decreased myosin heavy chain I (MHCI) mRNA levels previously observed in this condition [6] given their rapid induction in gastrocnemius muscle in response to exogenous T3 administration to hypothyroid [36,37] or fasted [38] rats. These events relate to the presence of a thyroid hormone response element (TRE) we identified in the rat UCP3 promoter [36] and a negative (n)TRE in the MHCI promoter as found in the functionally similar cardiac gene termed MHCβ [39]. Likewise, increased expression of PGC-1α in response to exercise in muscle has, although indirectly, been shown to rely on local T3 levels since deiodinase 2 ablation failed to result in exercise-induced PCG-1α expression in muscle [23]. The significantly increased T3 tissue levels we observed to be induced by mild endurance exercise with fasting would explain the further increase of PGC1-α expression with respect to fasting alone we observed previously [6]. Increased expression of PGC1-α, shown to directly control the expression of the ketolytic genes in skeletal muscle [40] was indeed associated with increased expression of BDH1 in gastrocnemius muscle in the combined intervention [6]. 

Increased BDNF-CREB signaling observed in this study provides one explanation for the previously observed boosted expression of IGF1R, UCP3, and PGC-1α during fasting in combination with mild endurance exercise [6]. Interestingly, IGF1R has been shown to be crucial for motor neuronal survival through the CREB-BDNF pathway in models of spinal muscle atrophy [41]. Transcription of UCP3 and PGC-1α is under the direct control of CREB since the promoters of both UCP3 and PGC-1α harbor CREB response elements [42,43]. 

Although under our experimental conditions BHB failed to accumulate in the prefrontal cortex we observed differential responses between interventions on the maturation of BDNF and activation of BDNF signaling. In the prefrontal cortex of fed rats, despite the absence of increased BHB, and in contrast to what has been observed in gastrocnemius muscle, exercise activated acute BDNF-Akt-mTORC1 signaling. Although this resulted in a reduction of deiodinase 3 mRNA levels, the lack of increase in FOXO 1 phosphorylation, associated with unaltered deiodinase 2 mRNA levels, may explain why T3 levels did not increase but reflected those measured in serum. The lack of increased tissue BHB and maturation of BDNF cannot explain the effect of short bouts of exercise on TRKB activation and downstream signaling we and others [7,8] observed in the prefrontal cortex. Apparently, a short bout of mild endurance exercise suffices to induce the release of BDNF known to be stored in the presynaptic terminal vesicles of excitatory neurons [44] and liberated upon synaptic excitation and calcium influx [44,45,46,47]. In view of the increases in serum BHB levels that we [6] and others [48,49] observed after 18 [48], 24 [49] and 66 h [6] of fasting in rodents, increased BHB levels, similar compared to that observed in gastrocnemius muscle, were expected in the prefrontal cortex. However, this was not observed, nor did BDNF signaling increase in response to fasting in the prefrontal cortex. In agreement, a 48 h fast did not result in increased levels of BDNF mRNA and protein in mouse brain [50], although longer-term exposure to ketones through alternate day fasting for a period of 3 months has been shown to increase BDNF protein levels in rat brain [51]. This may suggest that ketones produced by the liver during a 66 h-continuous fast, perhaps in contrast to a 3-month period of alternate fasting, are readily and completely used as fuel in the prefrontal cortex thus preventing their accumulation, hampering BDNF activation. Importantly, our observation regarding Akt-independent phosphorylation of P70S6K in prefrontal cortex, in association with increased leucine levels, provide an explanation as to why during fasting deiodinase 3 mRNA levels were downregulated and T3 was maintained at control levels, despite significantly decreased T3 serum levels. The increased prefrontal cortex T3 levels in response to mild endurance exercise during fasting with respect to controls can, in turn, be accounted for by the reduced deiodinase 3 mRNA levels associated with increased leucine levels and P70S6K activity, combined with an increased supply of T3 via the serum associated with increased production in gastrocnemius muscle. The increased T3 levels explain the expression of a selection of genes involved in neuronal regeneration known to be regulated by T3 in the prefrontal cortex [30,31]. Underlining these observations, deiodinase 3 ablation in the fetal brain has very recently been shown to increase the tissue’s sensitivity to T3 and the expression of genes including those studied here namely KLF9 and HR [52]. Taken together, although the mild endurance exercise bouts fail to activate prefrontal cortex BDNF signaling during fasting, they do activate T3-mediated events that are involved in the tissue’s maintenance.

## 5. Conclusions

This study brought to light differential gastrocnemius muscle and prefrontal cortex BDNF-Akt- mTOR signaling with a discriminatory role for BHB and BCAA tissue levels in response to mild endurance exercise during fasting. These events consistently lead to increased tissue thyroid hormone levels and together may contribute to understanding the physiological basis of the beneficial effects on muscle mass and strength as well as cognition associated with similar interventions. 

## Figures and Tables

**Figure 1 nutrients-14-01166-f001:**
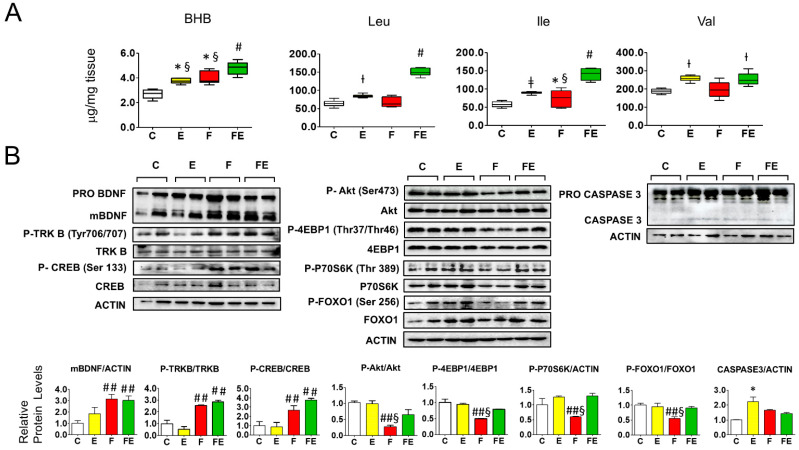
Association of beta hydroxy butyrate and branched-chain amino acids with BDNF-Akt-mTOR-FOXO1 signaling in gastrocnemius muscle of fasted rats undergoing mild endurance exercise. (**A**): Mass Spectrometry analysis of gastrocnemius muscle levels of BHB, Leu, Ile, and Val; (**B**): Western immunoblot analysis of gastrocnemius muscle proteins involved in BDNF-mTOR signaling, with quantified data shown below. Of note: P70S6K phosphorylation versus Actin ratios are depicted, the ratio with the non-phosphorylated protein did not vary. (**A**,**B**): Analysis was carried out comparing control rats with rats subjected to exercise, fasting, or fasting with exercise. (*N* = 4, *n* = 3). One way ANOVA was carried out followed by a Newman Keuls post-hoc test. * *p* < 0.05 vs. C, † *p* < 0.05 vs. C and F, ‡ *p* < 0.05 vs. C, F and FE, # *p* < 0.05 vs. E, F, and C, ## *p* < 0.05 vs. E and C, § *p* < 0.05 vs. FE. Abbreviations: N: biological replications, n: technical replications. BHB: beta hydroxy butyrate; Data are shown as means ± SEM.

**Figure 2 nutrients-14-01166-f002:**
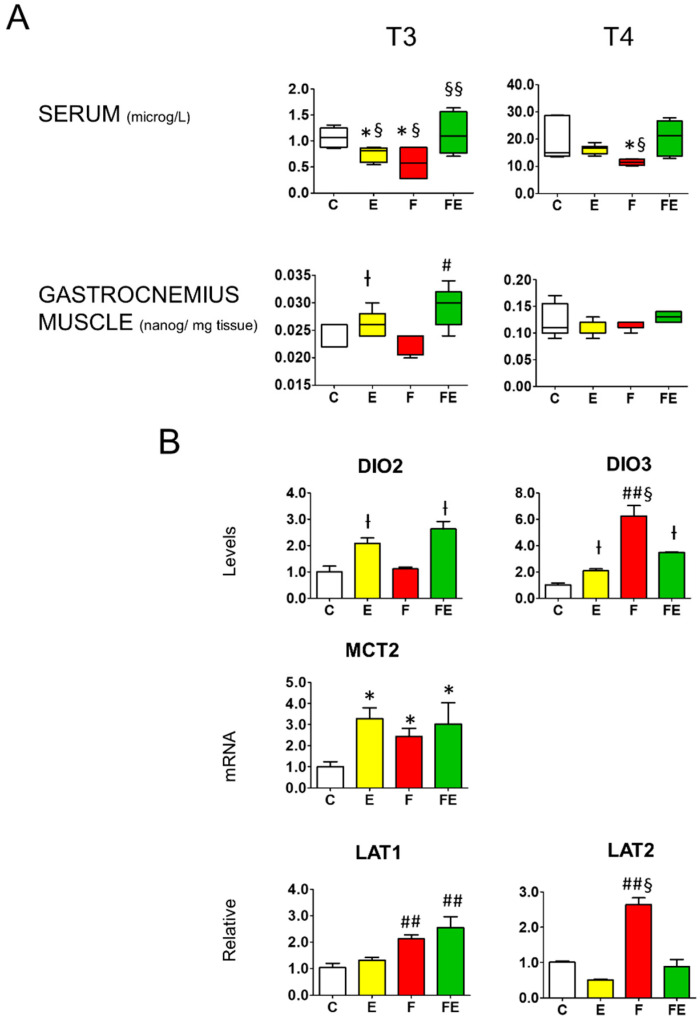
Comparison between serum and gastrocnemius muscle T3 and T4 levels and modulation of gastrocnemius muscle deiodinase 2 and deiodinase 3 mRNA levels, as well as those of BHB transporter MCT2, and amino acid transporters LAT1 and LAT2. (**A**): Mass Spectrometry analysis of serum and gastrocnemius muscle levels of T3 and T4. (**B**): Q-PCR analysis of deiodinase 2 and 3, MCT2, LAT1 and LAT2 mRNA levels in gastrocnemius muscle. (**A**,**B**): Analysis was carried out using control rats or rats subjected to exercise, fasting, or fasting with exercise (*N* = 4, *n* = 3). * *p* < 0.05 vs. C, † *p* < 0.05 vs. C and F, # *p* < 0.05 vs. E, F, and C, ## *p* < 0.05 vs. E and C, § *p* < 0.05 vs. FE, §§ *p* < 0.05 vs. F and E. One way ANOVA was carried out followed by a Newman Keuls post-hoc test. Data are shown as means ± SEM. Abbreviations: DIO2: deiodinase 2, DIO3: deiodinase 3, N: biological replications, n: technical replications.

**Figure 3 nutrients-14-01166-f003:**
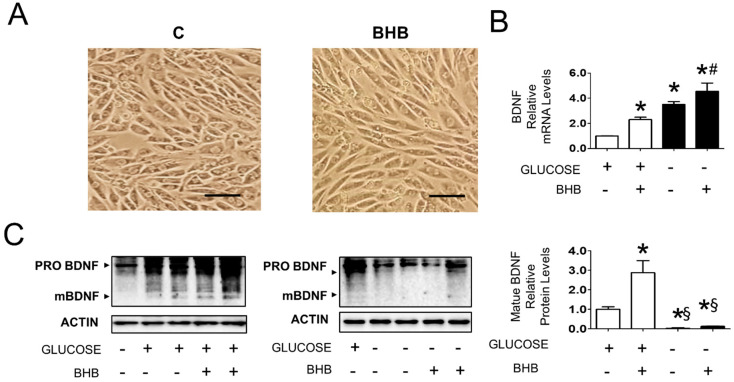
Effect of BHB on BDNF expression and maturation in L6 cells cultured in the presence and absence of 25 mM glucose (*N* = 6). (**A**): Morphology of cells cultured in absence or presence of 8 mM BHB for 3 days. The images represent cells cultured in absence of glucose, similar results were observed in presence of glucose (not shown). Scale bar: 100 μM. (**B**): BDNF mRNA levels in cells cultured in the presence and absence of 25 mM glucose and treated with solvent or 8 mM BHB. * *p* < 0.05 vs. control cells in presence of glucose, # *p* < 0.05 vs. control cells in absence of glucose; (**C**): Protein levels of proBDNF and mBDNF in cells cultured in presence and absence of 25 mM glucose and treated with solvent or 8 mM BHB. * *p* < 0.05 vs. control cells in presence of glucose, § *p* < 0.05 vs. cells in presence of glucose treated with solvent or 8 mM BHB. one way ANOVA was carried out followed by a Newman Keuls post-hoc test. Data are shown as means ± SEM.

**Figure 4 nutrients-14-01166-f004:**
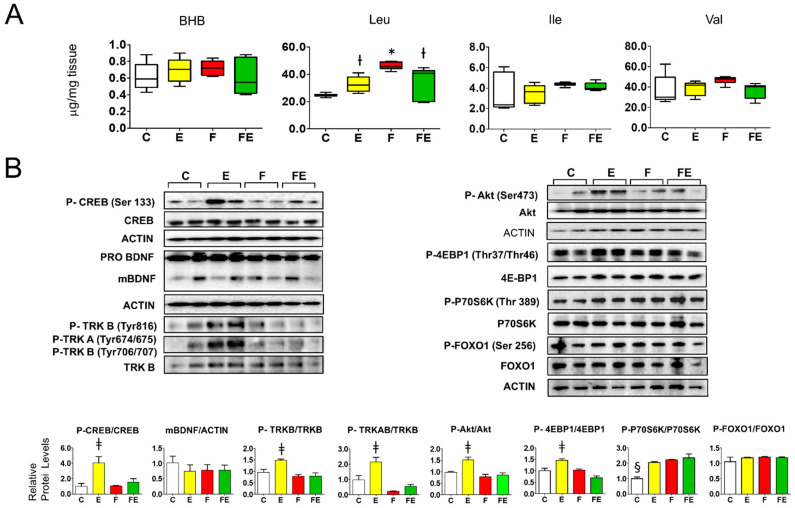
Association of tissue beta hydroxy butyrate and branched-chain amino acids with BDNF-Akt-mTOR-FOXO1 signaling in the prefrontal cortex of fasted rats undergoing mild endurance exercise. (**A**): Mass Spectrometry analysis of prefrontal cortex levels of BHB, Leu, Ile, and Val; (**B**): Western immunoblot analysis of prefrontal cortex proteins involved in BDNF-mTOR signaling, with quantified data shown below. (**A**,**B**): Analysis was carried out comparing control rats with rats subjected to exercise, fasting, or fasting with exercise. (*N* = 4, *n* = 3). One way ANOVA was carried out followed by a Newman Keuls post-hoc test. * *p* < 0.05 vs. C, † *p* < 0.05 vs. C and F, ‡ *p* < 0.05 vs. C, F and FE. § *p* < 0.05 vs. E, F and FE. Abbreviations: N: biological replications, n: technical replications. Data are shown as means ± SEM.

**Figure 5 nutrients-14-01166-f005:**
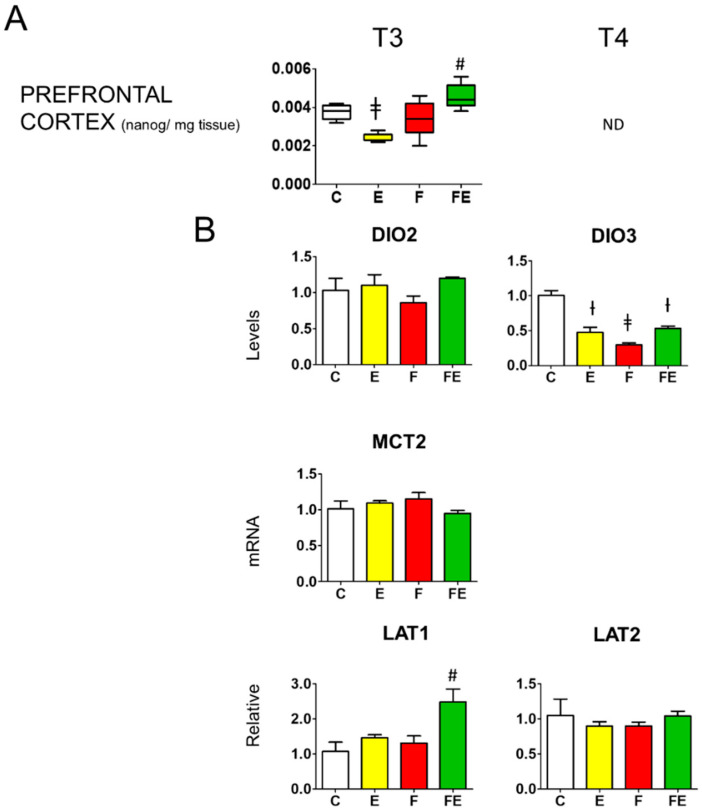
Prefrontal cortex T3 levels and modulation of prefrontal cortex deiodinase 2 and deiodinase 3 mRNA levels, as well as those of BHB transporter MCT2, and amino acid transporters LAT1 and LAT2. (**A**): Mass Spectrometry analysis prefrontal cortex levels of T3; (**B**): Q-PCR analysis of deiodinase 2 and 3, MCT2, LAT1, and LAT2 mRNA levels in the prefrontal cortex. (**A**,**B**): Analysis was carried out using control rats or rats subjected to exercise, fasting, or fasting with exercise (*N* = 4, *n* = 3). One way ANOVA was carried out followed by a Newman Keuls post-hoc test. † *p* < 0.05 vs. C and F, ‡ *p* < 0.05 vs. C, F and FE, # *p* < 0.05 vs. E, F, and C. Data are shown as means ± SEM. Abbreviations: DIO2: deiodinase 2, DIO3: deiodinase 3, N: biological replications, n: technical replications.

**Figure 6 nutrients-14-01166-f006:**
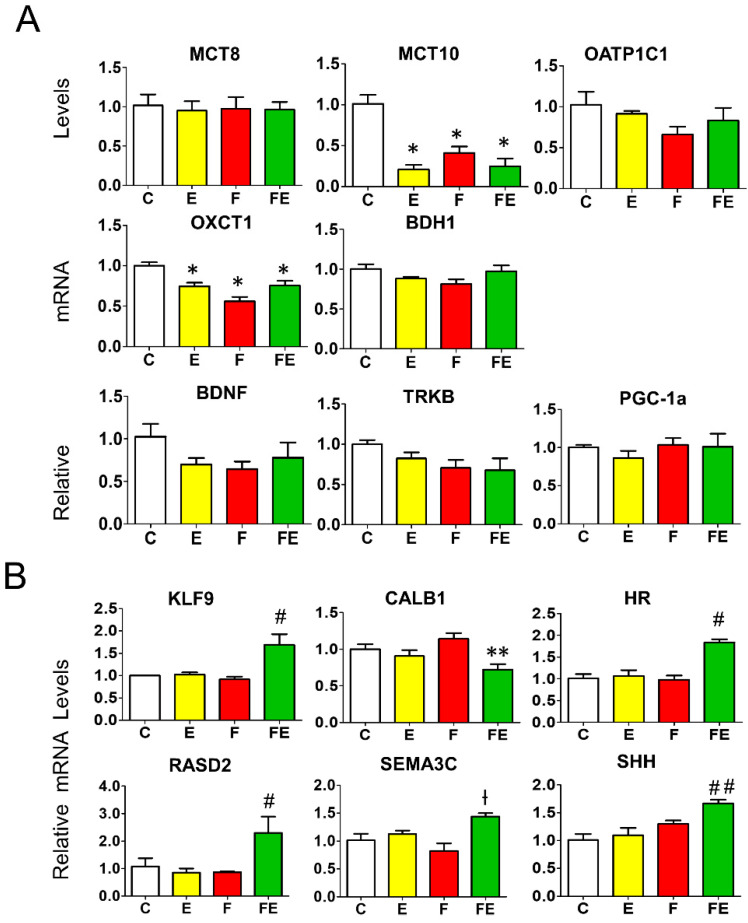
Expression of key genes involved in thyroid hormone and ketone action in the prefrontal cortex (**A**): Q-PCR analysis of genes involved in thyroid hormone transport, ketone transport and metabolism, and BDNF, TRKB and PGC-1α. (**B**): Q-PCR analysis of T3-responsive genes in the prefrontal cortex. (**A**,**B**): analysis was carried out using control rats or rats subjected to exercise, fasting, or fasting with exercise (*N* = 4, *n* = 3). One way ANOVA was carried out followed by a Newman Keuls post-hoc test. * *p* < 0.05 vs. C, ** *p* < 0.05 vs. F, † *p* < 0.05 vs. C and F, # *p* < 0.05 vs. E, F, and C, ## *p* < 0.05 vs. E and C. Abbreviations: N: biological replications, n: technical replications. Data are shown as means ± SEM.

**Table 1 nutrients-14-01166-t001:** List of precursor and product ions with collision energy and polarity measured by LCMSMS in this study.

Compound Name	Precursor Ion	Product Ion	Collision Energy	Polarity
T4	777.9	322.8	92	Positive
T4	777.9	731.6	28	Positive
T3	652	605.7	20	Positive
T3	652	197.1	84	Positive
BHB	105	64	5	Positive
BHB	105	59.1	10	Positive
Isoleucine Leucine	132.11	86	20	Positive
valine	118.1	55	20	Positive

BHB: beta hydroxy butyrate.

**Table 2 nutrients-14-01166-t002:** List of primers used for Q-PCR in this study.

Gene	Abbreviation	Forward Primer	Reverse Primer
3-Hydroxybutyrate dehydrogenase 1	BDH1	5′-TCTCGGACTGCCTACGCTAT-3′	5′-TAGAGGCTGGTGGCAGCTAT-3′
Brain-derived neurotrophic factor	BDNF	5′-GCCCAACGAAGAAAACCATA-3′	5′-CAAAGGCACTTGACTGCTGA-3′
Calbindin 1	CALB1	5′-TGTGGATCAATATGGGCAGA-3′	5′-ATCGAAAGAGCAGCAGGAAA-3′
Iodothyronine deiodinase 2	DIO2	5′-CAGTGAAGCGGAATGTCAGA-3′	5′-TTTCCCATTATCCCCTTTCC-3′
Iodothyronine deiodinase 3	DIO3	5′-ACAGATGAGCACAGCCACAG-3′	5′-CCAGAAAGCAAGCCAAAAAG-3′
Lysine demethylase and nuclear receptor corepressor	HR	5′-CTACAGCTCACCAGCATCCA -3′	5′-CCTCCCCAACTCCACAGTAA-3′
Kruppel-like factor 9	KLF9	5′-GGAAGATGCCACAATGGTTT-3′	5′-GATGTGATGCCATTCACGAG-3′
SLC7A5, L-type amino acid transporter 1	LAT1	5′-GTGAGGAGGCAGAGGTCAAG-3′	5′-CTGGGGACCTGAGTTCACAT-3′
SLC7A8, L-type amino acid transporter 2	LAT2	5′-GCTGGAAGAAGCCTGACATC-3′	5′-GCTGAAAATCAGCAGGAAGG-3′
SLC16A7, solute carrier family 16 member 7	MCT2	5′-ATCCGTCCACGAATCCAGTA-3′	5′-TGTGTAGGAATGGGCTAGGG-3′
SLC16A2, solute carrier family 16 member 2	MCT8	5′-ACAGCGCTTTCTGGTTCAGT-3′	5′-AAGGCCCAGATACGGTAGGT-3′
SLC16A10, solute carrier family 16 member 10	MCT10	5′-GTGCAATGGGTCTGTGTTTG-3′	5′-CCATGTTGTCATCGTCCTTG-3′
SLCO1C1, solute carrier organic anion transporter family member 1C1	OATP1C1	5′-CGAGGGATTTTCTTCCATCA-3′	5′-TGAACAGTGCTTGCACACAA-3′
3-oxoacid CoA transferase 1	OXCT1	5′-TGTGCAGCCATAGACTTTGC-3′	5′-GCACTCATGAAGCAAGACCA-3′
PPARG coactivator 1 alpha	PGC-1α	5′-GTCAACAGCAAAAGCCACAA-3′	5′-GTGTGAGGAGGGTCATCGTT-3′
RASD family, member 2	RASD2	5′-GCAAGAGCTCCATTGTCTCC-3′	5′-CGATGAAAGTCCTCGATGGT-3′
Semaphorin 3C	SEMA3C	5′-ATTTCGTCCGCGTTATTCAG -3′	5′-TTCCCCTGTTCAGGTAGGTG-3′
Sonic hedgehog	SHH	5′-CTGTACCACATTGGCACCTG-3′	5′-AGCTGGACTTGACTGCCATT-3′
Neurotrophic receptor tyrosine kinase 2	TRKB	5′-TTA GCC TCG TCAGGTGCTTT-3′	5′-TCCAGTCCAAACTGTGCTTG-3′
Beta actin	ACTIN	5′-TGTGTTGTCCCTGTATGCCT-3′	5′-CCCTCATAGATGGGCACAGT-3′

## Data Availability

The data presented in this study are available on request from the corresponding author.

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
