# Peer review of "Mild Endurance Exercise during Fasting Increases Gastrocnemius Muscle and Prefrontal Cortex Thyroid Hormone Levels through Differential BHB and BCAA-Mediated BDNF-mTOR Signaling in Rats"

_nutrients, 2022, doi:10.3390/nu14061166_

Round 1

Reviewer 1 Report

This is an interesting study of the effects of exercise and fasting on BDNF and hormone levels in the brain, muscle and serum of rats. There are several relatively minor things which should be addressed to improve the current manuscript.

  1. The Introduction would be easier to follow if it were split up into paragraphs for each topic covered. Many of the sentences are also quite long and difficult to follow. There are many sentences throughout the manuscript that have this problem also and would benefit from additional editing (one example is line 510-516).
  2. Section 2.2. has several issues. The description of the groups and number of rats/group is confusing. Perhaps it would be better to say "There were 4 experimental conditions used, each with a total of 4 rats used per condition", instead of "For each of a total of 4 experiments 4 groups of animals were selected. Each rat per group represented one separate experimental condition:" then follow with your description of each condition.
  3. For the exercise and fasting protocols I am first curious about the length of fasting used. Is 66 hours a common amount of time used? How long can a rat be fasted for safely? Did the exercise rats undergo 5 sessions in total? If the treadmill runs were done twice daily, did they complete only 1 session on the 1st or last day? How many hours were there between each treadmill run? I timeline might be beneficial to illustrate the protocol more clearly. In addition, was the room they did exercise at this temp also? Also, for the non-exercise rats, were they handled at all during the experiment, e.g. placed on inactive treadmill or just moved to training area without undergoing exercise at all? Finally, you have said twice that rats were sacrificed 4 hours after the last exercise bout, on both line 137 and line 138, one of these sentences can be removed.
  4. On lines 179 (".Data Processing:") and 251 ("Statistical analysis") there are statements  that look like they should be heading of a new section on a new line of the manuscript.
  5. The list of primers in section 2.5 might look better in a table. Make sure all of the gene names are defined as well.
  6. In the statistical analysis section of the methods, there appears to be a P missing from the statement " with  set at 0.05" on line 256. More importantly, you state that some analyses were done with Student's t-test. As all data appears to have been collected in all 4 of your experimental conditions, all data analysis should be done using ANOVA followed by post hoc testing as you have said the in next sentence.
  7. Related to this, in the results I think that the p values for all ANOVAs and post hoc tests should be presented, either in the text, or possibly as a table in supplementary material if adding it to the text would make the results section too long.
  8. In section 3.1 some sentences are very confusing to interpret which comparisons you are reporting. In the sentence "We observed the mature form of BDNF to be increased 2 fold 273 in the gastrocnemius muscle of 1 out of 4 controls and of 2 out of 4 chow-fed exercised rats, however mean induction was non-significant for both interventions" I'm not sure what the 1 out of 4 controls is being compared to? This sentence should be removed and you should just say there was no difference in mature BDNF in chow-fed exercise rats compared to controls. Similarly, the sentence "(TRKB) Tyr706/707 phosphorylation was not significantly induced in all control and chow-fed exercised animals, (Fig.1B, left panel) and neither was phosphorylation of CREB (Ser133) (Fig.1B left panel) and, of Akt (Ser473)" it would make more sense to say "(TRKB) Tyr706/707 phosphorylation was not significantly chow-fed exercised animals compared to controls (Fig.1B, left panel). Neither was phosphorylation of CREB (Ser133) (Fig.1B left panel) or of Akt (Ser473)."
  9. This also relates to the legend for Figure 1. The sentence about analysis should read "analysis was carried out comparing control animals with animals subjected to exercise, fasting, or fasting with exercise". Additionally in this figure legend, make sure the redefine the abbreviations used within the figure instead of just using the full names of BHB, Leu, Ile, Val. You should also explain why you present ratios of Western blot proteins here or in the methods.
  10. Line 318, "Instead," would be better as "In contrast,"
  11. The conclusion at the end of section 3.2 on lines 343-345 should be removed and confined to the Discussion with proper justification of how you have come to this conclusion.
  12. In Section 3.3 and Figure 3, I would have thought that the cells with no glucose and no BHB added would be the "control" group that all other groups should be compared to first. Therefore it would be shown first in all images on the figure, and all images should show the conditions in the same order. Is there a reason the protein levels in Figure 3C do not show examples from all conditions used?
  13. Throughout the Discussion, it is not always clear where you are writing about results of the current study or just discussing previous results not directly related to the current study, especially in the 2nd half of the 2nd paragraph. This may just require some further editing of the Discussion to ensure the importance of the current results are the main focus.

Author Response

Reviewer 1

We thank the reviewer for his/her comments.

“The Introduction would be easier to follow if it were split up into paragraphs for each topic covered. Many of the sentences are also quite long and difficult to follow. There are many sentences throughout the manuscript that have this problem also and would benefit from additional editing (one example is line 510-516).”

In accordance with the reviewer’s suggestion, we now arranged the introduction into paragraphs (without headings, which is not the journal’s style). We shortened the longer phrases, and highlighted them in yellow.

“Section 2.2. has several issues. The description of the groups and number of rats/group is confusing. Perhaps it would be better to say "There were 4 experimental conditions used, each with a total of 4 rats used per condition", instead of "For each of a total of 4 experiments 4 groups of animals were selected. Each rat per group represented one separate experimental condition:" then follow with your description of each condition.”

To render the description less confusing, we changed the text as the reviewer suggested (see lines 133-138)

“For the exercise and fasting protocols I am first curious about the length of fasting used. Is 66 hours a common amount of time used? How long can a rat be fasted for safely?”

We are pleased to define the issue the reviewer brings to attention. Commonly used is 24- 48 h but 72h is also applied in the literature (see for example PMC6831946). According to Boston University “Complete food deprivation of 72 hours in rats and 48 hours in mice is acceptable with scientific justification.  Research staff responsible for monitoring animals on food regulation studies must be trained and competent to evaluate the animal's condition”(see for link https://www.bu.edu/researchsupport/compliance/animal-care/working-with-animals/food-regulation-and-restriction-in rodents).

“Did the exercise rats undergo 5 sessions in total? If the treadmill runs were done twice daily, did they complete only 1 session on the 1st or last day? How many hours were there between each treadmill run? I timeline might be beneficial to illustrate the protocol more clearly. “

The rats were subjected to 5 sessions in total, on the third day only 1 session was carried out before sacrifice. A timeline of the experiment has been published in a previous study (6). To meet the reviewer’s request, we refer to his/her comment  with a new sentence in line 144-145.

“(In addition, was the room they did exercise at this temp also? Also, for the non-exercise rats, were they handled at all during the experiment, e.g. placed on inactive treadmill or just moved to training area without undergoing exercise at all? Finally, you have said twice that rats were sacrificed 4 hours after the last exercise bout, on both line 137 and line 138, one of these sentences can be removed.”

All animals were familiarized with the treadmill,  initial environmental temperature was set at 25°C and constantly monitored to ensure that the temperature inside the plexiglass cover did not exceed 28°C during the exercise sessions. This has been described in the methods section of our previous work (6) and has been better specified  now  in the methods section (see lines 140-143). In addition, we removed the first sentence related to the sacrifice time point.

“On lines 179 (".Data Processing:") and 251 ("Statistical analysis") there are statements  that look like they should be heading of a new section on a new line of the manuscript.”

We agree with the reviewer and have now placed the headings on a new line. (see lines 188 and 224)

“The list of primers in section 2.5 might look better in a table. Make sure all of the gene names are defined as well.”

In accordance with the reviewer, we have inserted a table, Table 2, including the gene names.

“In the statistical analysis section of the methods, there appears to be a P missing from the statement " with  set at 0.05" on line 256. More importantly, you state that some analyses were done with Student's t-test. As all data appears to have been collected in all 4 of your experimental conditions, all data analysis should be done using ANOVA followed by post hoc testing as you have said the in next sentence.”

We apologize to the reviewer, the Student’s T test remained in the manuscript by error due to a previous version, we solely performed ANOVA analysis throughout. In accordance with the reviewer, we have changed the text inserting the alpha symbol setting the border for the P-value (see lines 229-237).

“Related to this, in the results I think that the p values for all ANOVAs and post hoc tests should be presented, either in the text, or possibly as a table in supplementary material if adding it to the text would make the results section too long.”

In accordance with the reviewer’s request, with regard to the ANOVAs, we have inserted the exact P values next to all measured parameters in supplementary Table S1.We have specified the ANOVA test in the legends of each figure. For the post hoc test (Student’s Newman Keuls) we have set the P value throughout at 0.05.

“In section 3.1 some sentences are very confusing to interpret which comparisons you are reporting. In the sentence "We observed the mature form of BDNF to be increased 2 fold 273 in the gastrocnemius muscle of 1 out of 4 controls and of 2 out of 4 chow-fed exercised rats, however mean induction was non-significant for both interventions" I'm not sure what the 1 out of 4 controls is being compared to? This sentence should be removed and you should just say there was no difference in mature BDNF in chow-fed exercise rats compared to controls. Similarly, the sentence "(TRKB) Tyr706/707 phosphorylation was not significantly induced in all control and chow-fed exercised animals, (Fig.1B, left panel) and neither was phosphorylation of CREB (Ser133) (Fig.1B left panel) and, of Akt (Ser473)" it would make more sense to say "(TRKB) Tyr706/707 phosphorylation was not significantly chow-fed exercised animals compared to controls (Fig.1B, left panel). Neither was phosphorylation of CREB (Ser133) (Fig.1B left panel) or of Akt (Ser473)."

To make the description in section 3.1 more clear, we have inserted the suggested changes (see lines 245-249). We thank the reviewer for his/her efforts.

“This also relates to the legend for Figure 1. The sentence about analysis should read "analysis was carried out comparing control animals with animals subjected to exercise, fasting, or fasting with exercise". Additionally in this figure legend, make sure the redefine the abbreviations used within the figure instead of just using the full names of BHB, Leu, Ile, Val. You should also explain why you present ratios of Western blot proteins here or in the methods.”

We have changed the legend of figure 1 in accordance with the reviewer’s suggestions (see lines 278-280) and also of figure 4, where the same changes needed to be made (see lines 368-375). We also explained why we calculated the phospho-total ratios in the methods section stating: “The ratio of the phosphorylated versus total proteins was calculated to assess the phosphorylation status.” (see lines 222-223).

“Line 318, "Instead," would be better as "In contrast,"”

In accordance with the reviewer, we have changed the sentence. (line 288)

“The conclusion at the end of section 3.2 on lines 343-345 should be removed and confined to the Discussion with proper justification of how you have come to this conclusion.”

In accordance with the reviewer, we removed this phrase from section 3.2. as this is explained in the discussion.

“In Section 3.3 and Figure 3, I would have thought that the cells with no glucose and no BHB added would be the "control" group that all other groups should be compared to first. Therefore it would be shown first in all images on the figure, and all images should show the conditions in the same order. Is there a reason the protein levels in Figure 3C do not show examples from all conditions used?”

We would like to clarify that presence of glucose would represent the chow fed control groups, and absence of glucose would represent the fasted group, as explained in the results section 3.3. To underline our logic, in section 3.3, we have changed the phrase related to this by starting with the glucose  condition and ending with the glucose deprived condition (line  320-323)  All conditions are presented on the blots in the figure. The first lane of both panels  serves to compare the signal intensity with and without glucose on the same blot, and to be able to directly compare the two separate blots based on the intensity of the signals. The next 4  lanes on the blots show BDNF protein in presence of glucose (representing fed controls) (left panel), or in absence of glucose (representing fasting) (middle panel) in absence or presence of BHB.  The histogram, following the concept that cells incubated in presence of glucose represent the fed controls, initiates with data obtained in presence of glucose.  In sharp contrast to the fasting situation in gastrocnemius muscle, in absence of glucose, BDNF maturation is not detectable, either or not in presence of BHB (middle panel). We hope that the current version is satisfactory for the reviewer.

“Throughout the Discussion, it is not always clear where you are writing about results of the current study or just discussing previous results not directly related to the current study, especially in the 2nd half of the 2nd paragraph. This may just require some further editing of the Discussion to ensure the importance of the current results are the main focus.”

We agree with the reviewer and have attempted to bring the present data more into focus. (see yellow highlighted parts).

Reviewer 2 Report

This manuscript provides information about the effect of mild endurance exercise during fasting on BHB and BCAA and its signaling pathways in the gastrocnemius muscle and prefrontal cortex. This research topic has been studied in various models, including human studies. Hence, the new findings in this manuscript are not surprising. The experimental design is not diverse since the authors repeated the same experiments using different tissues. More importantly, there are too many grammatical errors and typos, making it hard for readers to understand the manuscripts. Statistics are incorrectly used for all figures. Fig 3A has not been described in the text, and Figure 3 is unnecessary for this manuscript. 

Author Response

Reviewer 2

We thank the reviewer for his/her comments.

“This manuscript provides information about the effect of mild endurance exercise during fasting on BHB and BCAA and its signaling pathways in the gastrocnemius muscle and prefrontal cortex. This research topic has been studied in various models, including human studies. Hence, the new findings in this manuscript are not surprising.”

We are aware of the extensive research on this topic. This is the first work to study the relationship between central and peripheral BDNF-Akt-mTORC1axis and BHB levels and their synergism with systemic and tissue thyroid hormone signaling. The study provides insights on the effects of exercise and fasting on peripheral thyroid hormone modulation and confirms the critical role of muscle-specific modulation of thyroid hormone in pathophysiological conditions. Tissue BHB and BCAA and T3  levels have not been measured in other studies and  this study reveals that mild exercise with fasting increases all these compounds in muscle but only leucine and T3 in the prefrontal cortex.

“The experimental design is not diverse since the authors repeated the same experiments using different tissues.”

To meet the reviewer’s consideration we would like to underlne that the experimental design is not diverse since we aimed to study whether exercise during fasting induced BDNF-mTOR signaling and whether this would lead to increased intracellular T3 levels and action in both cortex and gastrocnemius muscle. As stated in the Conclusion,  our results show differential gastrocnemius muscle and prefrontal cortex BDNF-Akt- mTOR signaling with a discriminatory role for BHB and BCAA tissue levels in response to mild endurance exercise during fasting. These events consistently lead to increased tissue thyroid hormone levels and together may contribute to understanding the physiological basis of the beneficial effects on muscle mass and strength as well as cognition associated with similar interventions.

“More importantly, there are too many grammatical errors and typos, making it hard for readers to understand the manuscripts”

We acknowledge the reviewer’s comment and we carefully proofread the manuscript and corrected for grammatical errors (highlighted in yellow in the typeset manuscript)  

“Statistics are incorrectly used for all figures.”

We apologize to the reviewer, the “student’s T-test analysis”  has been left in the text by error due to a previous version. All analyses have been performed with 1 way ANOVA, and the differences have been indicated with the appropriate symbols as indicated in the legends.  Also, in accordance with reviewer 1, we have now  inserted the ANOVA P values in the supplemental table S1.

“Fig 3A has not been described in the text, and Figure 3 is unnecessary for this manuscript.”

With regard to Fig3A, we inserted the phrase highlighted in yellow “Cell morphology was unaltered following treatment with BHB (Fig.3A).”(line 324). The use of the muscle cell model is an effort  to verify whether  BHB directly stimulates BDNF in muscle cells as it does in cortex neuronal cells (see reference 32). Data in Figure 3 report that BHB stimulated BDNF transcription and maturation hence supporting a functional association. We hope the reviewer will reconsider his/her suggestion.

Round 2

Reviewer 2 Report

Authors still need to improve and be careful over the manuscript. For example, they must change words on line 371, "gastrocnemius muscle" to "prefrontal cortex".

Author Response

We thank the reviewer for his/her comments and have now changed the words on line 371, "gastrocnemius muscle" to "prefrontal cortex". In addition, we corrected the text throughout, all changes are highlighted in yellow. We hope that the revised manuscript is now to the reviewer's satisfaction.
